# FSVFG: Towards Immersive Full-Scene Volumetric Video Streaming with Adaptive Feature Grid

## ABSTRACT

Full-scene volumetric video streaming, an emerging technology providing immersive viewing experiences via the Internet, is receiving increasing attention from both the academic and industrial communities. Considering the vast amount of full-scene volumetric data to be streamed and the limited bandwidth on the internet, achieving adaptive full-scene volumetric video streaming over the internet presents a significant challenge. Inspired by the advantages offered by neural fields, especially the feature grid method, we propose FSVFG, a novel full-scene volumetric video streaming system integrated feature grids as the representation of volumetric content. FSVFG employs an incremental training approach for feature grids and stores the features and residuals between adjacent grids as frames. To support adaptive streaming, we delve into the data structure and rendering processes of feature grids and propose bandwidth adaptation mechanisms. The mechanisms involve a coarse ray-marching for the selection of features and residuals to be sent, and achieve variable bitrate streaming by Level-of-Detail (LoD) and residual filtering. Based on these mechanisms, FSVFG achieves adaptive streaming by adaptively balancing the transmission of feature and residual according to the available bandwidth. Our preliminary results demonstrate the effectiveness of FSVFG, demonstrating its ability to improve visual quality and reduce bandwidth requirements of full-scene volumetric video streaming.

## CCS CONCEPTS

• **Information systems** → **Multimedia streaming**; • **Networks** → *Application layer protocols*.

## 1 INTRODUCTION

Volumetric video (VV) captures content in 3D, providing viewers with a six-degree-of-freedom (6DoF) motion. Recently, volumetric video has been at the forefront of immersive media and holds immense potential for revolutionizing sectors such as entertainment, gaming, education, and telecommunications. Since the volumetric video content is represented by dense explicit geometric structure, e.g., point cloud (PtCl), streaming volumetric video over the Internet consumes huge bandwidth consumption. Recent studies have proposed solutions to mitigate data usage, such as leveraging tile-based viewport, occlusion, and distance visibility [6, 11, 43]. Most of them focus on streaming individual objects, e.g., a single person.

*MM'24, Oct 28-Nov 1, Melbourne, Australia*
© 2024 Association for Computing Machinery.
ACM ISBN 978-x-xxxx-xxxx-x/YY/MM...$15.00
https://doi.org/10.1145/nnnnnnn.nnnnnnn

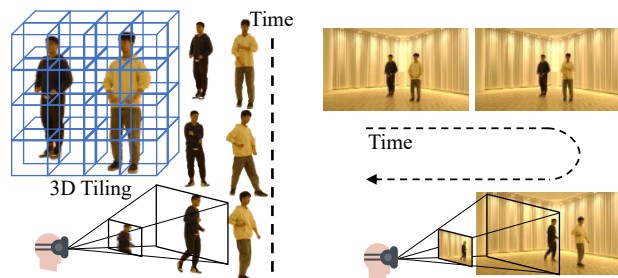

(a) Object-based video     (b) Full-scene volumetric video

**Figure 1: Illustrations of the object-based and full-scene VV**

In contrast, full-scene volumetric video comprises both 3D objects and the entire environment, leading to a more engaging and interactive experience than the object-based volumetric video. As is shown in Figure 1, full-scene volumetric video typically has a significantly larger but low-dynamic environment and relatively small areas with high-dynamic objects. The huge difference in object sizes poses a significant challenge for adaptive streaming [7].

Neural fields have demonstrated powerful capabilities in viewing quality with a compact model, offering new opportunities in full-scene volumetric video streaming. Neural fields such as NeRF store the volumetric content in a parameterized neural network. This network takes coordinates as inputs and outputs color and density, which can then be used in volume rendering. These methods have proven their ability to enhance visual quality while reducing storage costs compared to explicit representations. Recent studies have extended neural fields by incorporating time as an additional input to represent volumetric videos [33, 34]. However, the complexity of spatial changes in dynamic 3D scenes necessitates the use of numerous parameters in the neural network for accurate representation [28, 29], posing a substantial challenge to real-time rendering [20] and often resulting in compromised rendering quality. The use of numerous parameters also enlarges the data volume and makes it too large to stream, especially for full-scene volumetric content.

To achieve faster rendering and improve visual quality, recent studies have proposed the feature grid [26, 37] that stores a series of trainable positional encodings in a grid of features for each region in the scene. During inference, the input coordinates are first replaced by related encodings in the grids before being fed into the decoder, a single-layer MLP that outputs color and density. With fast query on grids and low complexity of the decoder, the feature grid can achieve better quality than other neural fields while maintaining real-time rendering performance [17]. In the field of streaming, most current feature grid streaming studies focus on streaming a single moving object [19, 34, 41, 45]. When considering the full scene, the environment data size significantly outweighs that of the moving objects [7], making these streaming methods impractical.

To this end, we propose FSVFG, an adaptive full-scene volumetric video streaming system that integrates feature grids as the representation of volumetric content. Our design and implementation of FSVFG pose two technical challenges.

**How to avoid massive data transfer?** Full-scene volumetric video typically has a relatively large but low-dynamic environment with relatively small areas containing high-dynamic objects. Therefore, there is a strong inter-frame similarity among full-scene volumetric video frames. Inter-frame similarity is widely utilized to reduce the size of video frames. It allows for the storage and streaming of only the residuals in dynamic regions, thereby significantly reducing the bandwidth consumption of streaming. However, leveraging it within a feature grid presents a non-trivial challenge. Due to the high degree of freedom among feature grid parameters, it is difficult to maintain similarity between adjacent frames in the training process [42]. To address this issue, we propose inter-frame regularized training that exploits the inter-frame similarity of volumetric video frames by applying a regularization term in incremental training [42] to the feature grid.

**How to support bandwidth adaptation?** To support bandwidth adaptation, the streaming system should have a bitrate ladder that trades off quality and bandwidth to deal with fluctuating bandwidth. The traditional tile-based adaptive point cloud streaming [6] is implemented by controlling the density of tiled point clouds for network adaptation. However, in feature grid rendering, the density of the network does not necessarily represent the trade-off between quality and transmission. Therefore, they cannot achieve satisfactory performance in the streaming of feature grids. To tackle this problem, we take an insight into the rendering process and propose to select features and residuals to be sent according to a coarse ray-marching. We also delve into the data structure of the feature grid and achieve variable bitrate streaming by LoD (Level-of-Detail) and residual filtering. Based on these mechanisms, we achieve a heuristic algorithm to select the LoD of features and change the value of the threshold.

We have implemented FSVFG based on InstantNGP and conducted extensive experiments on typical datasets and comparisons with baselines. The key contributions are summarized as follows:

**(1)** To the best of our knowledge, FSVFG is a full-scene video streaming system integrated feature grid. By leveraging the ability of the feature grid in adaptive streaming, it reduces the bandwidth and optimizes conventional transmission.

**(2)** Based on the insight into the unique characteristics of the feature grid data structure and rendering process, we tackle these problems and design adaptive streaming mechanisms that improve visual quality and reduce network bandwidth requirements.

**(3)** We develop a prototype to evaluate the effectiveness of our approach. Our evaluation of typical datasets and comparisons with baselines demonstrate promising results.

## 2 BACKGROUND AND MOTIVATION

### 2.1 Volumetric Video Streaming

Volumetric video streaming has drawn substantial research attention in recent years. Current streaming systems can divided into two scopes. Some studies such as V-PCC [31] codec or Vues [22] project volumetric videos into 2D videos before streaming, while others stream the volumetric content directly. Our work falls within the latter scope.

Some previous studies in this scope have investigated volumetric video streaming based on point cloud and have achieved adaptive streaming [1, 35, 36, 40]. ViVo [6] was a pioneer in tiling-based visibility-aware adaptive volumetric video streaming, and other studies like GROOT [11], AITransfer [8], and YuZu [47] introduced different technologies to optimize tiling-based adaptive streaming. More recently, CaV3 [18] and Hermes [43] utilized inter-frame similarity of volumetric video frames to decrease bandwidth consumption of adaptive streaming, and MetaStream [5] integrates several innovations into a comprehensive system to accomplish volumetric video live streaming. However, with the huge amount of attributes (i.e. position and color) in the point cloud and other explicit representations, these methods suffer from large data volumes [17, 20] even with the use of well-designed encoders [39, 43].

Recently, innovative explicit representation methods such as 3D Gaussian splatting [10, 44, 46] and Plenoxels [4, 13] are proposed, but they are far from practical for networked applications due to their additional attribute requirements, such as spherical harmonic coefficients, which pose significant challenges for compression.

### 2.2 Neural Field

Neural fields such as NeRF [25] approximate the continuous signal in 3D space via a parametric continuous function that takes coordinates as inputs and outputs attribute vectors. These methods have proven their ability to enhance visual quality while reducing storage costs compared to explicit representations [3, 14]. Besides, they also eliminate the need for depth cameras in 3D reconstruction, thereby significantly reducing its costs and making it accessible to a wider audience. As such, these methods exhibit considerable potential for volumetric video streaming and have attracted significant research attention [9]. Unfortunately, despite the growing interest in streamable neural fields [37], most current research primarily focuses on compression [3, 13, 41], but not considering practical issues on volumetric video streaming such as visibility and bandwidth adaptation. To address this gap, our paper aims to leverage the benefits of neural fields to facilitate adaptive full-scene volumetric video streaming.

Within the large community of neural field research, many types of neural fields can represent volumetric video. The selection of neural field representation is critical for achieving adaptive volumetric video streaming and should consider several factors. Essentially, the chosen representation should deliver high quality and be capable of real-time rendering on client devices. Besides, it should be divisible temporally and spatially into smaller segments to achieve visibility-aware adaptation [6]. Furthermore, it should be capable of partitioning into different quality levels of varying sizes to accommodate varying bandwidth requirements for transmission [18, 47].

### 2.3 Feature Grid

Through investigation, we found that the feature grid can meet the above requirements [17]. Feature grid [2, 21, 27] is a special class of neural fields that is renowned in the research community for its superior quality and the ability to render at interactive rates. As is shown in Figure 2, a typical feature grid method comprises

a feature grid, an MLP, and global Spherical Harmonics. Given a viewport as input, the feature grid method operates as follows [24, 38]: **(1)** Ray marching: sampling points along the ray $\vec{r} = \vec{o} + t\vec{d}$ from the target image and find the containing voxels in the feature grid of each point. **(2)** Positional embedding: the embedding vector of a point is linearly interpolated based on its relative position within the containing voxel in different resolutions. **(3)** Directional embedding: direction $\vec{d}$ of the ray is transformed into an embedding vector by Spherical Harmonics. **(4)** Inference: the positional and directional embedding vectors of each sample point are fed into the MLP, outputting the color and density. **(5)** Volume rendering: the color and density of the sample points are accumulated to derive the pixel color on the target image.

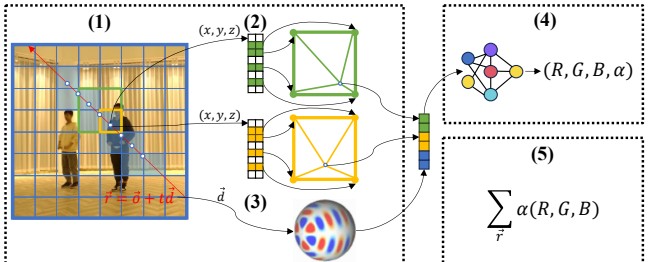

**Figure 2: Illustration of the feature grid method**

By embedding coordinates into a high-dimensional space, the feature grid method shifts the complexity of the signal representation from the MLP to the feature grid, thereby achieving superior quality and rendering performance, and its grid structure facilitates spatial splitting. In recent studies, Liu et al. [17] have preliminarily explored achieving adaptive streaming by exploiting the separability of the feature grid, they did not consider inter-frame similarity to reduce bandwidth consumption. Liao et al. [41] have leveraged inter-frame similarity for frame compression based on feature grids, but they did not consider adaptive streaming and did not perform well for long videos. In summary, the feature grid is suitable for adaptive volumetric video streaming, but no practical system based on the feature grid exists.

To this end, this paper proposes achieving adaptive volumetric video streaming based on the feature grid. Specifically, we mainly use NGP [26] that stores the features in a compact table in our implementation and experiments, as this method represents the current mainstream of feature grid studies. As such, our innovations and methods could be extended to most other feature grid methods such as dense grid [37, 38], VQAD [23, 37] or VQRF [12].

## 3 SYSTEM OVERVIEW

The FSVFG system is designed to train a sequence of feature grids for volumetric video representation, and streams video-on-demand volumetric content represented by these feature grids from an Internet server to the client. The workflow of the FSVFG system consists of an offline training phase and an online adaptation phase, as is shown in Figure 3. In the offline phase, a scene is captured by a synchronized camera array from various perspectives. The captured data are then used to train a series of feature grids by inter-frame regularized training (§4). After the training process, each feature grid is partitioned into different Levels of Detail (LoD) and stored

separately. In the online phase, the FSVFG server receives the predicted viewport from the client and sends the features or residuals selectively (§5.1). Then the selected features are filtered by LoD and inter-frame residuals are filtered by threshold according to the network condition, thereby achieving adaptation (§5.2).

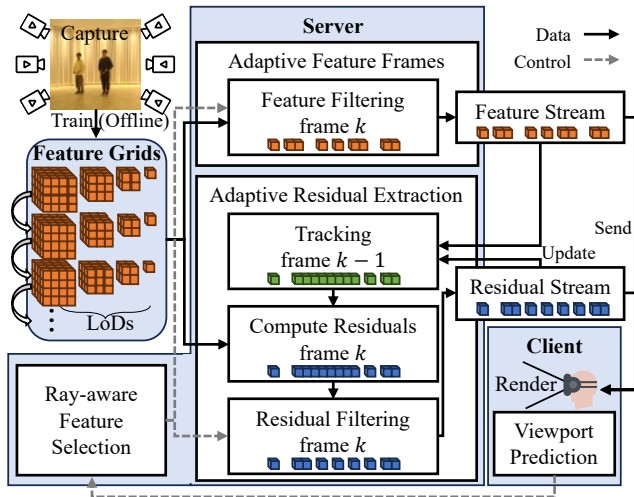

**Figure 3: Overview of ifNGP streaming system**

## 4 INTER-FRAME REGULARIZED TRAINING

The concept of neural field incremental training [42] stems from the understanding of 3D scenes. Unlike 2D videos, there is little "camera movement" in a 3D case. Those environment elements such as floors, walls, or buildings are unlikely to demonstrate high dynamics throughout a volumetric video [34]. Therefore, the volumetric video frames are well-aligned and inter-frame residuals most exist in those regions containing moving objects. This well-aligned inter-frame similarity can be leveraged to reduce the storage size of the frame and the bandwidth consumption of the streaming, without the need for motion estimation and compensation for alignment.

To leverage inter-frame similarity, incremental training loads the neural field parameters of the previous frame at the beginning of the training and freeze some parameters in the neural field during the training [13]. Ideally, the training process should only make minor adjustments to the parameters, and the inter-frame residual can be extracted and stored, which would be more space-efficient than storing the parameters of the neural fields.

However, the naive application of incremental training to the feature grid does not ensure minor changes to the parameters. In the feature grid, the parameters lack correlation with each other. In this case, some parameters in static areas may change during the incremental training process, leading to relatively larger residuals.

To address this issue, we propose to add a regularization term that minimizes the residual between the parameters of the current and previous frames in the loss function. The modified loss function $\mathcal{L}(\theta)$ is represented by Equation 1.

$$\mathcal{L}(\theta) = \mathcal{L}_0(\theta) + \lambda \sum_{i \in N} \theta_{i,k} - \theta_{i,k-1} \tag{1}$$

where $\mathcal{L}_0(\theta)$ denotes the original loss function (e.g. rendering loss [25, 42]), $\lambda$ represents the weight of the regularization, $\theta_{i,k}$ represents one of the parameters in the feature grid of frame $k$, and $N$ is the total numbers of the parameters in the feature grid. This regularization term can be incorporated into the loss function of any type of neural field, including feature grids, and can significantly reduce the volume of residuals that need to be stored, leading to improved space efficiency and less bandwidth consumption of streaming.

In practice, using frames captured simultaneously from different perspectives by the synchronized camera array [48] as training data, our inter-frame regularized training process operates as follows: for the initial frame, we train the feature grid without freezing and regularization. For each subsequent frame, we load the parameters of the previous frame and update only the parameters of the feature grid with the regularization term in the loss function. After training each frame, the parameters are saved for the training of the next frame. Through this process, we can sequentially train the feature grids for each frame, resulting in a sequence of files containing parameters of feature grids, each representing a frame of the volumetric video.

## 5 ONLINE ADAPTATION

### 5.1 Ray-aware Feature Selection

To achieve adaptation in volumetric video streaming systems, the common strategy is to partition a frame into tiles and stream the tiles within the viewport [6, 11]. However, for the feature grid, the density of the points does not necessarily represent the trade-off between quality and transmission. The rendering process of feature grids is significantly different from traditional explicit volumetric video representations such as point clouds. These traditional representations are typically rendered through rasterization or splatting that projects 3D objects into a raster image. In contrast, feature grids are rendered by ray marching, where each pixel on the output image corresponds to a ray, and each ray is traced within the grid. The pixel color is then calculated using the features of the voxels along the ray (Figure 2). Therefore, unlike in rasterization where each element in the received tiles contributes to the output image, in ray marching, only the features along the ray contribute to the output image. As a result, packaging feature grids in tiles for transmission could lead to the transmission of redundant features that do not contribute to the output image (Figure 4). Moreover, compact representations [26, 37] are widely adopted in storage-efficient feature grids such as VQAD [23, 37] or VQRF [12]. In these feature grids, one feature may be used in different positions more than once. Thus, existing spatially tiling methods would cause a single feature to appear in multiple tiles, causing transmission redundancy.

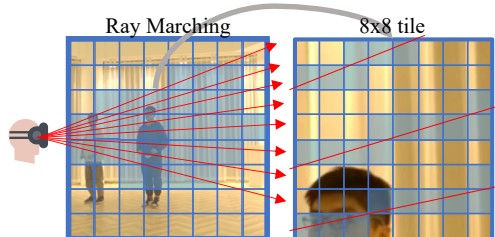

**Figure 4: Illustration of ray marching and tiling**

Considering the characteristics of feature grid rendering, we propose the Ray-aware Feature Selection mechanism that selects and sends only those features along the ray to the client. In practice, after the training process, the scene is divided into coarse cubes (i.e., $128^3$ cubes), and sampling is performed in each cube to compute the density value inside it. The highest and lowest density values of each cube are then stored in a density grid. During the streaming process, our ray-aware adaptation component receives the predicted viewports from the client and performs a coarse ray marching (Figure 5) on the server.

The step size is set to match the ray marching in the client-side rendering process. Each step corresponds to a sampling point along the ray, and features of the voxels containing this point are selected, and their index is added to a set $V_k$. For empty regions (highest density=0), we apply the empty space skipping method [15] to skip the sampling point to the next occupied region (highest density>0). The output of ray-aware feature selection is a set of indices $V_k$, which will be used in the subsequent feature filtering and residual filtering processes.

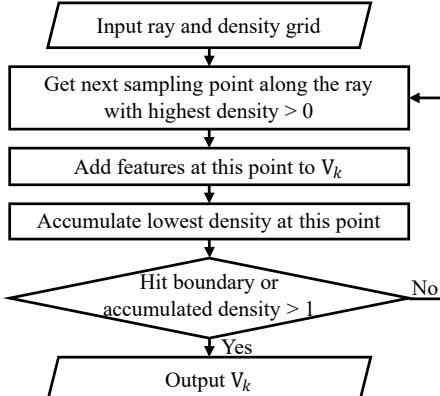

**Figure 5: Illustration of Ray-aware Feature Selection**

Since this process only involves query and accumulation operations without any time-consuming rendering processes (e.g., MLP inference), it can keep pace with the video playback frame rate when run on GPU devices on the server.

### 5.2 Variable Bitrate

Adaptive streaming aims to trade off video quality and bandwidth consumption under fluctuating network conditions. In full-scene volumetric video streaming, the adaptation process should also incorporate the progressive refinement of the environment due to its substantial size and the inherent difficulty in transmitting it entirely to the client at the start of video playback. Related works include Fumos [16] that integrate neural compression and octree-based coding to achieve progressive refinement for continuous point cloud video streaming. Similarly, LiveVV [7] enables adaptation by using different Point Cloud Density Levels (PDL) for full-scene volumetric video streaming. However, as mentioned before, the density of the points does not necessarily represent the trade-off between quality and transmission for the feature grid. Therefore, layer split methods based on density are not applicable to feature grid streaming.

Besides, existing adaptation methods in volumetric video streaming do not take into account the dependencies between frames as

they do not employ inter-frame correlation [30, 43] for compression. When the scene is represented by feature grids, we can leverage inter-frame correlation by computing and streaming the residuals of the features from adjacent frames for low-dynamic regions to reduce bandwidth consumption. Despite the benefits, sending both features and residuals in a stream can introduce complexities in adaptive streaming, e.g. how to achieve variable bitrate on residuals and how to balance the streaming of features and residuals.

The goal of this section is to design mechanisms to deal with such complexities and support adaptive streaming.

*5.2.1 Adaptive Feature Frames.* To enable adaptation in streaming features, we divide the feature grid into different quality levels using level-of-detail (LoD), which is a characteristic provided by multiresolution encoding in most feature grid implementations [26, 37, 38]. In our feature stream, LoD can provide adaptation through an adjustable value $L_k$ for the frame $k$, where only the features in those LoDs lower than this $L_k$ will be sent to the user. For instance, at the beginning of the video, a lower $L_k$ can be set to stream the scene, allowing the user to quickly achieve an immersive experience, then progressively increasing the $L_k$ to refine the quality during playback.

---

**Algorithm 1** Feature Filtering

---

**Input:** parameters of feature grid $\{\theta_{i,k}\}$, results of ray-aware feature selection $V_k$, set of index $S_{k-1}$ of parameters that have been sent to the client, limit $M$ on the number of parameters

**Output:** filtered features $F_k$, the updated set of index $S_k$

1: **function** FEAT_FILTER($\{\theta_{i,k}\}, V_k, S_{k-1}, M$)
2:     $F_k^{all} \leftarrow \{i | i \in V_k$ and $i \notin S_{k-1}$ and $\theta_{i,k} \neq 0\}$;
3:     $C_k \leftarrow |\text{LoD}_1 \cap F_k^{all}|; L_k \leftarrow 1; l \leftarrow 2$;
4:     **while** $C_k + |\text{LoD}_l \cap F_k^{all}| \leq M$ and $l \leq L$ **do**
5:         $C_k \leftarrow C_k + |\text{LoD}_l \cap F_k^{all}|$;
6:         $L_k \leftarrow l; l \leftarrow l + 1$;
7:     **end while**
8:     $F_k \leftarrow \{\theta_{i,k} | i \in \text{LoD}_l \cap F_k^{all}$ and $l \leq L_k\}$;
9:     **if** $M - |F_k| > 0$ and $L_k < L$ **then**
10:        $F_k^{rand} \leftarrow \text{RandomSelect}(\text{LoD}_{L_k+1} \cap F_k^{all}, M - |F_k|)$;
11:        $F_k \leftarrow F_k \cup \{\theta_{i,k} | i \in F_k^{rand}\}$;
12:     **end if**
13:     $S_k \leftarrow S_{k-1} \cup \{i | \theta_{i,k} \in F_k\}$
14:     **return** $F_k, S_k$;
15: **end function**

---

The feature filtering process is outlined in Algorithm 1. In essence, feature filtering gathers those selected parameters in $V_k$ that have not been sent to the client, from lower LoD levels to higher LoD levels, until the number of parameters reaches the given limit $M$. If the parameters in the gathered layers do not fully utilize the given limit, we further fill it by selecting parameters randomly from a higher level.

*5.2.2 Adaptive Residual Extraction.* To enable adaptation in streaming residuals, an intuitive method is to store the residuals of different LoDs as separate quality levels. However, in adaptive streaming, after switching from a higher level to a lower level, the transmission of residuals for the higher level ceases. As a result, features in the

higher level remain constant, while the features in the lower level keep changing, leading to a mismatch of features across different LoDs and degrading the visual quality. Therefore, simply applying LoD in both feature and residual streaming is not advisable.

Based on the observation that larger residuals have a greater impact on visual quality, we divide the residual grid into different quality levels by their numerical value, thereby avoiding the quality degradation caused by mismatched features. Specifically, a threshold $T_k$ is chosen adaptively for residual filtering for the frame $k$, and the residual $\Delta\theta_{i,k}$ for parameter $\theta_{i,k}$ of the features is computed according to Equation 2:

$$
\begin{aligned}
\Delta\theta_{i,k} &= f(\theta_{i,k} - \hat{\theta}_{i,k-1}, T_k) \\
\hat{\theta}_{i,k} &= \Delta\theta_{i,k} + \hat{\theta}_{i,k-1} \\
\Delta\theta_{i,1} &= f(\theta_{i,1} - \theta_{i,0}, T_1) \\
\hat{\theta}_{i,1} &= \Delta\theta_{i,1} + \theta_{i,0} \\
f(x, t) &= \begin{cases} x & |x| \geq t \\ 0 & |x| < t \end{cases}
\end{aligned}
\tag{2}
$$

where $\Delta\theta_{i,k}$ represents the residual of parameter $i$ in frame $k$, while $\theta_{i,k}$ and $\hat{\theta}_{i,k}$ represents true value and tracked value of the parameter respectively. Therefore, during streaming, adjusting $T_k$ can control the number of residuals to be sent.

Note that the feature grid and the residuals have been quantized to float16 (16-bit floating point) to reduce bandwidth consumption, which can cause quantization errors. In Equation 2, the tracked value $\hat{\theta}_{i,k-1}$ represents what the parameters would be after adding the residual, and residuals computed based on these tracked values would reset the error, preventing the errors from accumulating.

---

**Algorithm 2** Residual Filtering

---

**Input:** parameters of feature grid $\{\theta_{i,k}\}$ and tracked values $\{\hat{\theta}_{i,k-1}\}$, results of ray-aware feature selection $V_k$, set of index $S_{k-1}$ of parameters that have been sent to the client, limit $M$ on the number of parameters, lower bound $T_m$ of residuals

**Output:** filtered residuals $R_k$, updated tracked values $\{\hat{\theta}_{i,k}\}$

1: **function** RES_FILTER($\{\theta_{i,k}\}, \{\hat{\theta}_{i,k-1}\}, V_k, S_{k-1}, M, T_m$)
2:     $R_k^{all} \leftarrow V_k \cap S_{k-1}$;
3:     $T_k \leftarrow \max(\text{TopK}(\{|\theta_{i,k} - \hat{\theta}_{i,k}|| i \in R_k^{all}\}, M), T_m)$;
4:     **for** $i \in R_k^{all}$ **do**
5:         $\Delta\theta_{i,k}, \hat{\theta}_{i,k} \leftarrow \text{ComputeResidual}(\theta_{i,k}, \hat{\theta}_{i,k-1}, T_k)$; ▷ Eq(2)
6:     **end for**
7:     $R_k \leftarrow \{\Delta\theta_{i,k} | i \in R_k^{all}$ and $\Delta\theta_{i,k} \neq 0\}$;
8:     **for** $i \notin R_k^{all}$ **do**
9:         $\hat{\theta}_{i,k} \leftarrow \hat{\theta}_{i,k-1}$
10:     **end for**
11:     **return** $R_k, \{\hat{\theta}_{i,k}\}$;
12: **end function**

---

The residual filtering process is described in Algorithm 2. Given a limit $M$ on the number of residuals, residual filtering first identifies the $M$th largest residual using an efficient TopK algorithm on the GPU [32]. Considering that the small values have little effect on the visual quality, we set a lower bound $T_m$ and take the larger value between $M$th largest residual and $T_m$ as the threshold $T_k$. Finally,

with the threshold $T_k$, the filtered residuals are computed according to Equation 2.

*5.2.3 Rate Allocation.* As visual quality improves with the transmission of more features and residuals, maximizing visual quality equates to maximizing the use of available bandwidth. In this context, the goal of our allocation is to fully utilize the bandwidth to transmit as many features and residuals as possible.

Based on this idea, we provide a bandwidth constraint $B_k$ for the frame $k$ to control the bandwidth consumption. Assuming that the compression ratio $r$ can be estimated, and the frame rate $f$ is fixed, the capacity of parameters or residuals for each frame can be computed as $M = \frac{B_k}{fr}$. The adaptation decision is then to allocate the $M$ to the residuals and features of each frame. We provide another control parameter $\gamma \in [0, 1]$ for the rate allocation, fill the $\gamma M$ by feature filtering and fill the rest part by residual filtering. Considering that in some cases the number of residuals may be few, we perform a reallocation of features after the residual filtering.

---

**Algorithm 3** Rate Allocation at frame $k$

---

**Input:** frame rate $f$, bandwidth limitation $B_k$ and compression ratio $r$, bitrate allocation factor $\gamma$, viewport $\text{Cam}_k$ uploaded by the client

**Output:** The set of features $F_k$ and residuals $R_k$ to be sent to client

1: $M \leftarrow \frac{B_k}{fr}$
2: Loading parameters $\{\theta_{i,k}\}$ and density grid for frame $k$
3: $V_k \leftarrow \text{FeatureSelect}(\text{Cam}_k)$;  ▷ Ray-aware Feature Selection
4: $F_k, S_k \leftarrow \text{FEAT\_FILTER}(\{\theta_{i,k}\}, V_k, S_{k-1}, \gamma M)$;  ▷ Alg.1
5: $R_k, \{\hat{\theta}_{i,k}\} \leftarrow \text{RES\_FILTER}(\{\theta_{i,k}\}, V_k, S_{k-1}, M - |F_k|, T_m)$;▷ Alg.2
6: **if** $M - |R_k| - |F_k| > 0$ and $L_k < L$ **then**
7:     $F_k, S_k \leftarrow \text{FEAT\_FILTER}(\{\theta_{i,k}\}, V_k, S_{k-1}, M - |R_k|)$;  ▷ Alg.1
8: **end if**
9: Save $S_k$ and $\{\hat{\theta}_{i,k}\}$ for next frame;

---

The rate allocation algorithm is described in Algorithm 3. With these algorithms, the bandwidth is filled with features in low LoD in the first frame. Then, in the early period of streaming, the bandwidth is allocated among features and residuals, progressively refining the visual quality while ensuring the smooth movement of the object in the scene. After a period of streaming, most of the features will have been sent to the client, and the streaming is now mainly filled by residuals. In this process, the control parameters $B_k$ and $\gamma$ can be used to trade off the quality and bandwidth consumption. Note that our allocation algorithms are highly parallelizable and can be executed on the server at high frame rates.

In a nutshell, our system provides parameters such as $M$ and $\gamma$ to trade off visual quality and bandwidth consumption, meeting the necessary conditions for adaptive streaming strategies. Existing adaptation algorithms, such as Lyapunov optimization [16] or multiarmed bandit [18] perform well and can be applied within this context. The specific design of these algorithms is not the primary focus of this study.

# 6 EVALUATION

According to the methods introduced above, we build a streaming system and evaluate FSVFG at a system level. Our dataset includes 9 stereo videos from 3 datasets: "taekwondo", "walking" from the

st-nerf dataset [48], "coffee martini", "flame steak", "sear steak" from the Neural 3D Video Synthesis dataset [14], and "discussion", "step in", "trimming", "VR headset" from the Meet Room dataset [13]. We use the Instant-NGP [26] system as our testing platform, with the feature grid size set to $2^{19}$, and the number of Levels of Detail (LoDs) fixed at 8.

## 6.1 System-Level Evaluation

*6.1.1 Performance of Inter-frame Regularized Training.* To demonstrate the performance of inter-frame regularized training, we compared the video quality under different bandwidth constraints for models trained with and without inter-frame regularization. The training configurations are the same as Instant-NGP. For comparison, we train two sequences of feature grids for each video in the dataset using the incremental training and inter-frame regularized training process introduced in Section 4. The feature grid for the initial frame is trained for 30,000 steps, while feature grids for subsequent frames are trained for 10,000 steps. After training, the features are saved into files, and residuals are extracted for each frame and stored individually as binary files. Then for each frame, we set different thresholds to filter and compress the residuals as Equation 2, adjusting the threshold so that the size of the compressed frame data ranged from 1 to 6MB. For each threshold configuration, we obtain the rendering results of viewports in the dataset and assess the PSNR (Peak Signal-to-Noise Ratio) of the rendering results against the ground truth.

Figure 6 illustrates the impact of varying thresholds on visual quality and frame size across different videos. It is evident that as the frame size increases (threshold of residual filtering decreases), incremental training without inter-frame regularization can achieve higher quality. However, as the frame size gradually decreases, the video quality deteriorates more severely, and the quality fluctuation is also greater in some videos. This is because the parameters within the feature grid are less interdependent compared to NeRF, giving them higher degrees of freedom and more considerable fluctuations during training. As a result, the incremental training designed for NeRF cannot effectively control the parameter changes within a smaller range, leading to larger inter-frame residuals, even in static regions. Therefore, during residual filtering, to achieve the same frame size, a larger threshold needs to be set, which filters out the residuals belonging to the moving areas, causing a decline in quality. Furthermore, since a larger threshold is set, the parameter errors must accumulate to higher values before residuals are used for compensation, which introduces greater instability to the video quality.

*6.1.2 Performance of Online Adaptation.* To demonstrate the performance of our ray-aware feature selection and variable bitrate mechanisms, we build a prototype system and evaluate it. For comparison, we use a traditional tile-based method as the baseline. In the baseline scenario, the parameters of the features and residuals are clustered by the size of 4096 ($16^3$) as tiles in the baseline, with different levels of detail (LoD) clustered independently. A cluster is sent whenever a ray passes through any feature within it. We play the volumetric video and collect data on viewport movement using a Quest 3 device. Our adaptation mechanisms and the baseline are run on the collected viewport movement. We set the bandwidth

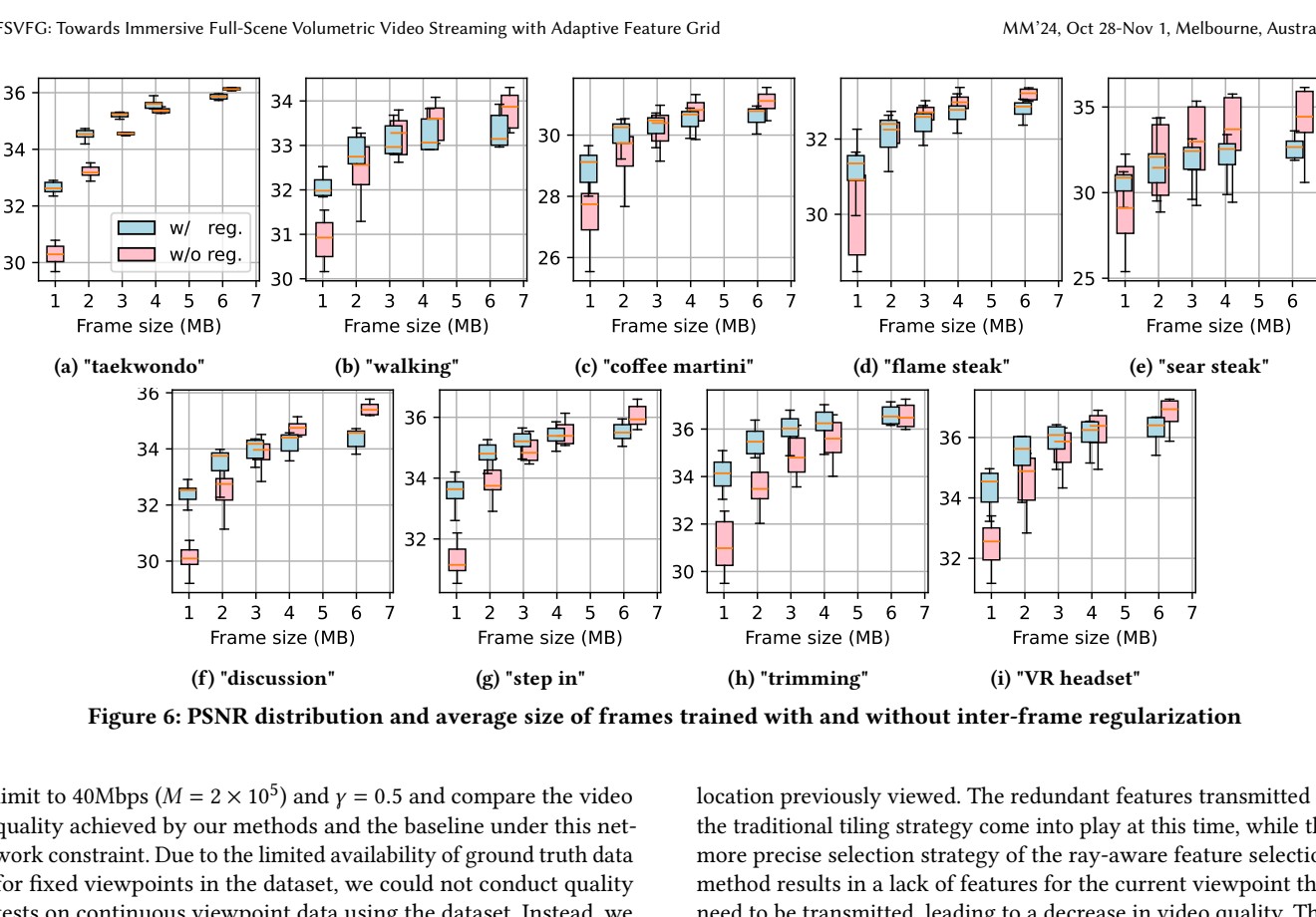

**Figure 6: PSNR distribution and average size of frames trained with and without inter-frame regularization**

limit to 40Mbps ($M = 2 \times 10^5$) and $\gamma = 0.5$ and compare the video quality achieved by our methods and the baseline under this network constraint. Due to the limited availability of ground truth data for fixed viewpoints in the dataset, we could not conduct quality tests on continuous viewpoint data using the dataset. Instead, we use the output of the original feature grid trained on each frame under the corresponding viewpoint as the ground truth, and the PSNR is calculated.

Our experiment results is shown in Figure 7. Under bandwidth constraints, our ray-aware feature selection method generally outperforms the traditional tiling strategy. Especially at the beginning of the video, our method allows the video quality to rise more quickly and stabilize. This is because, in comparison with the traditional tiling strategy, our ray-aware feature selection method can transmit those features that contribute to the output image more precisely. The traditional tiling strategy transmits a large number of redundant features that do not contribute to the output image. To achieve the same video quality, the traditional tiling strategy needs to transmit more features, and features at lower LoDs require more time to complete transmission before higher LoD features can be transmitted. Therefore, under the same bandwidth constraints, their video quality is lower. Additionally, our ray-aware feature selection method can ensure relatively stable video quality when the user's viewpoint observes high-dynamic areas, e.g., "taekwondo" frames 60-80, "trimming" frames 80-100. This is because our method saves the bandwidth required for feature transmission, allowing more bandwidth to be used for transmitting residuals, thereby achieving higher video quality in those high-dynamic areas. When video quality tends to stabilize, our method can sometimes cause instability in video quality in a short period, e.g., "coffee martini" frames 60-80, "sear steak" frames 60-80. This is because, in these video frames, the user's viewpoint moves quickly and just happens to move back to a

location previously viewed. The redundant features transmitted in the traditional tiling strategy come into play at this time, while the more precise selection strategy of the ray-aware feature selection method results in a lack of features for the current viewpoint that need to be transmitted, leading to a decrease in video quality. This phenomenon can be further studied in future research.

*6.1.3 Performance of Runtime.* To evaluate the performance of our FSVFG runtime, we run our prototype system on PCs with Intel i7-12700 CPU and NVIDIA GeForce RTX 3080 8G GPU. Table 1 shows the average frame rate of frame loading and rendering, demonstrating that our implementation can support real-time playback experience. As the frame loading depends on the viewport data received from the client side, its frame rate is inherently constrained by the rendering rate. When the frame rate escalates to a higher range (above 30 fps), it is observable that the frame loading rate lags behind the rendering rate. Profiling indicates that the primary bottleneck resides in the disk read speed for frame retrieval.

**Table 1: Average frame rate of rendering and frame loading process**

| Video | Process | 1080p | 720p | 540p |
|---|---|---|---|---|
| "coffee martini" | rendering | 12.0 | 20.3 | 34.5 |
| | frame loading | 12.0 | 20.3 | 28.3 |
| "trimming" | rendering | 12.1 | 22.4 | 35.6 |
| | frame loading | 12.1 | 22.4 | 27.3 |

## 6.2 Case study

We conduct a case study on "VR headset" to present the detailed rate allocation and visual samples of our FSVFG system. Figure 8 illustrates the timeline in the FSVFG system when streaming "VR headset" with $M = 2 \times 10^5$ and $\gamma = 0.5$. In first 10 frames, the first

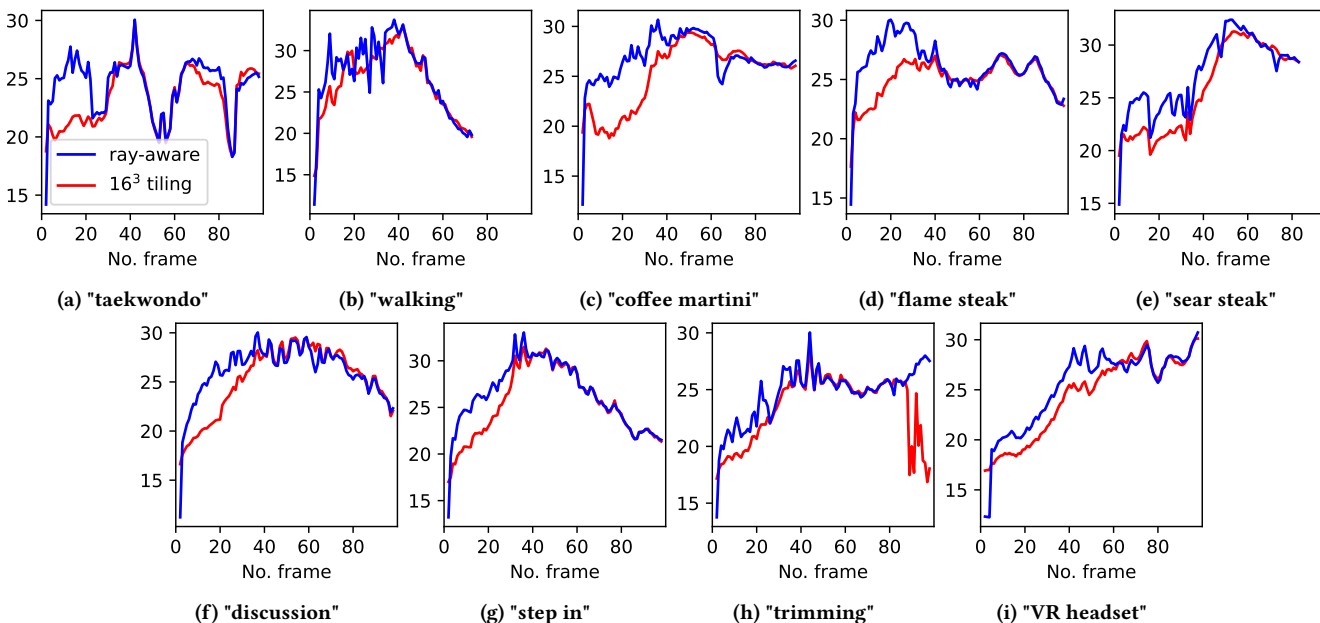

(a) "taekwondo"  (b) "walking"  (c) "coffee martini"  (d) "flame steak"  (e) "sear steak"

(f) "discussion"  (g) "step in"  (h) "trimming"  (i) "VR headset"

**Figure 7: Comparison of PSNR fluctuations in volumetric video between our adaptation methods and the tiling-based method with a bandwidth limit of 40Mbps**

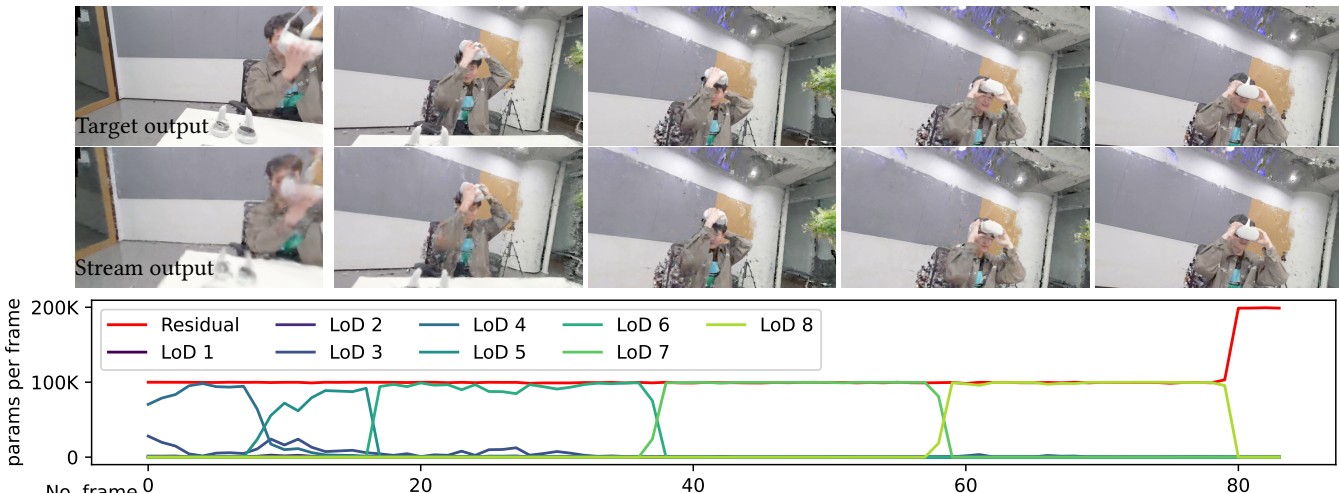

**Figure 8: Case study: A timeline of visual samples and number of parameters sent for each frame (Video: "VR heasdet")**

few LoDs are downloaded, offering the user a blurry outline of the video content. As the streaming progresses through the subsequent frames, the transmission of data at lower LoDs is largely completed. The subsequent arrival of features from higher LoDs progressively refines the video quality. In the output visual samples, the scene gradually becomes clearer and more detailed as the streaming process continues. At the 80th frame, features from all the LoDs have fully arrived. From this point onwards, all bandwidth resources are dedicated to the transmission of residual data. As a result, the video quality stabilizes, maintaining a consistent level of clarity and detail for the remainder of the video stream.

## 7 CONCLUSION

This paper introduced the FSVFG system to enable adaptive volumetric video streaming based on feature grids. The proposed method utilized incremental frozen training to enable effective compression of frame size and achieve rate adaptation by ray-aware feature selection and filtering. Experimental results show that our system outperforms existing methods including incremental training and tile-based adaptation. By integrating our implementation into high-level applications, we can further develop practical implementations of an implicit volumetric video streaming system for real-world deployment.

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
