# OpenReview forum: "FSVFG: Towards Immersive Full-Scene Volumetric Video Streaming with Adaptive Feature Grid"
_acmmm.org/ACMMM/2024/Conference — MM2024 Poster_

### Official Review · Reviewer_9Fnb · 2024-05-21

**Rating:** 4
**Confidence:** 3

**Summary:**

This paper proposes a new technique based on the Instant-NGP model [1] in the neural field, aiming to apply it to volumetric video streaming. The author utilized an incremental learning technique inspired by inter-frame similarity. Moreover, they proposed a level-of-detail (LoD) and rate allocation technique to support various bandwidth levels, which is based on ray-aware feature selection. Finally, they present the rendering results of effectively applying Instant-NGP to volumetric video streaming.

**Strengths:**

Overall, this paper is well-written and easy to follow. The motivation is also clear, and it contains several insights for deploying the neural field method to the adaptive streaming framework. The proposed technique is very well explained in detail with a specific algorithm table.

In addition, this paper proposes a mechanism similar to viewport rendering using ray-aware feature selection. This approach, which calculates the density value of the neural field model, appears to have potential for application in most adaptive volumetric streaming. Using filtering methods to transmit the optimal representation with limited resources, such as the number of parameters or bitrate, seems reasonable.

**Limitations:**

There is major concern as follows:

The comparison with experimental results or analysis is not robust, leaving unanswered questions about design choices. The modified loss function for Inter-reg in the paper doesn not seems to be novel, although it is substantial for video representation learning. If so, the proposed algorithms and system should be considered as the main contributions. Unfortunately, there is no provided ablation study corresponding to algorithms 1, 2, and 3.

There is also a concern that the only comparison method in Figure 7 involves a single 16^3 tiling. This only demonstrates that ray-aware feature selection outperforms the conventional method, without clarifying the contribution of each proposed technique to this improvement.

Because of this, these feature filtering algorithms somewhat looks ad-hoc decision. How can it be demonstrated that these algorithms optimize parameters for LoD properly? For example, we could consider introducing an importance score as in VQRF, or consider magnitude-based pruning methods [2,3]. Unfortunately, the lack of key experimental results makes it difficult to understand how much the proposed techniques have improved. Experimental results based on the filtering threshold value should be presented and compared to straight-forward pruning techniques.

--

I believe the equation (1) should be <$\mathcal{L}(\theta)=\mathcal{L}_0(\theta)+\lambda \sum_{i \in N} |\theta_{i, k}-\theta_{i, k-1}|$>

[1] Müller, Thomas, et al. "Instant neural graphics primitives with a multiresolution hash encoding." ACM transactions on graphics (TOG) 41.4 (2022): 1-15.

[2] Li, Lingzhi, et al. "Compressing volumetric radiance fields to 1 mb." Proceedings of the IEEE/CVF Conference on Computer Vision and Pattern Recognition. 2023.

[3] Lee, Jaeho, et al. "Layer-adaptive sparsity for the magnitude-based pruning." arXiv preprint arXiv:2010.07611 (2020).

**Suitability:**

3

---

### Official Review · Reviewer_gcxQ · 2024-05-24

**Rating:** 4
**Confidence:** 1

**Summary:**

This paper introduces FSVFG, a system that improves full-scene volumetric video streaming using feature grids. Volumetric videos, offering immersive 3D experiences, typically require high bandwidth. FSVFG tackles this by using incremental training and adaptive streaming mechanisms, adjusting data transmission based on network conditions.

**Strengths:**

This approach is innovative because it shifts the representation of volumetric content from traditional point clouds or explicit geometric structures to feature grids, which are more efficient for both storage and streaming;
The paper includes extensive experimental evaluations that demonstrate the effectiveness of the FSVFG system.

**Limitations:**

The details of the LoD adjustment mechanism might not be thoroughly explained;
Since Gaussian splatting has become an important method in 3D vision, please provide more details regarding why it is far from practical for networked applications due to additional attribute requirements;
While the encoding and transmission stages are optimized separately, there is a lack of end-to-end joint optimization across the entire system, potentially missing the globally optimal solution.

**Suitability:**

3

---

### Official Review · Reviewer_U3MK · 2024-05-24

**Rating:** 3
**Confidence:** 3

**Summary:**

This paper presented a 3D tiling method over the 3d volume for adaptive volumetric video streaming.

**Strengths:**

1. tiling strategy is a popular method for any kind of bandwidth consuming media streaming system, it's very reasonable to use it for 6-DoF video
2. this paper gives enough experiment results for visual quality and frame size. I can see the method they proposed is solid.

**Limitations:**

1. a lot of related works and methods are mentioned in section 2.1, but no one appears in evaluation. I know some of them are not able to be compared. but at least one, as a baseline, so we can know your current position. If more comparison can be done, this paper will be more complete and competitive.

**Suitability:**

3

---

### Official Review · Reviewer_cW3S · 2024-06-04

**Rating:** 5
**Confidence:** 3

**Summary:**

This paper presents a novel full-scene volumetric video streaming system that uses feature grids to represent volumetric content and achieve adaptive streaming over the internet. Differently from other works in this area, it leverages nerf representations to perform streaming

**Strengths:**

The paper is interesting since it leverages concept typical of the adaptive streaming domain, to volumetric videos represented by nerfs. Results suggests that this approach can work better in terms of adaption. The evaluation performed given a good initial sense of the performance of the solution

**Limitations:**

-	Not sure if the regularization component of the loss to better represent the parameters of the network can be claimed as a contribution? It seems like a smart but obvious extension of the current loss definition
-	Figure 3 is complicated and hard to interpret, maybe it can be made bigger?
-	It is unclear to what extent the comparison with tiling approaches allows to draw conclusions on which method is better. A bit more context should be provided
-	Sections 5.2.1 and 5.2.2 should be expanded a bit to give more details on the rationale of the choices. As of now, it is hard to understand the intuition behind the approach.

**Suitability:**

3

---

### Meta-Review · Area_Chair_LChu · 2024-07-03

**Recommendation:** Accept (Poster)
**Confidence:** 4

**Metareview:**

The paper is systematic and detailed, clearly written.
The different steps of the algorithm are described in detail.
The average rating after rebuttal has slightly improved from 4.0 to 4.25. The rebuttal letter is considered as rather good and constructive. Some space is missed by too frequent appreciation of each comment. In turn, the summarization of concerns and joint addressing of these are good points.
Most points of criticism are well addressed in the rebuttal. It is recommended to indeed include all mentioned improvements also in a possible camera ready (CR) paper.
In particular, the usage of feature grids / neural fields / nerfs representations (R1, 3), or as R4 puts it, the Instant-NGP model (see ref. in R4's review), represent a relevant contribution, especially in the area of adaptive volumetric video streaming (all reviewers). Here, aspects from adaptive streaming and nerf-based volumetric video are combined, indicating advantages in adaptation (R1).
A wider-scale impact is mentioned by R4: "This approach, which calculates the density value of the neural field model, appears to have potential for application in most adaptive volumetric streaming."
Several reviewers positively mention the evaluations carried out (R1, 2, 3). In turn, R4 sees limitations with the evaluations, e.g., regarding robustness. In the rebuttal, some additional comparisons and ablations are provided, which should also be included in the CR paper.
The further limitations of the experiments pointed out by R4 are not fully resolved in rebuttal and unlikely to be feasible to fully be addressed.
Similarly, from my own perspective, using PSNR as only evaluation metric for "visual quality" is certainly a strong limitation, even if this corresponds to often used practice. As it does not optimally represent visual perception, and certainly not for volumetric video, this aspect should at least be mentioned, in addition to a better consideration of the limitations with the evaluation section mentioned especially by R4, and somewhat by R1, 2. Also with regard to feature filtering (R4), the considerations in the rebuttal should be included in a possible CR paper version.
The complementary thoughts on Gaussian splatting in response to R3 are interesting and could be included to some extent in the paper, too.

As the overall approach appears quite novel in the specific domain of adaptive streaming, and the overall rating is clearly above 4, I recommend acceptance. I propose acceptance as poster to enable a more fine-grained explanation of the algorithm details, but can also imagine an oral presentation, if space allows, in comparison also to possibly even more highly ranked papers.